# Efficient bulk mass accommodation and dissociation of $N_2O_5$ in neutral aqueous aerosol

Goran Gržinić [1,2], Thorsten Bartels-Rausch[1], Andreas Türler[2,3], Markus Ammann[2]

[1] Laboratory of Environmental Chemistry, Paul Scherrer Institut, Villigen, 5232, Switzerland
[2] Department of Chemistry and Biochemistry, University of Bern, Bern, 3012, Switzerland
[3] Laboratory of Radiochemistry, Paul Scherrer Institut, Villigen, 5232, Switzerland

*Correspondence to*: Markus Ammann (markus.ammann@psi.ch)

**Abstract.** An isotope exchange experiment with the short-lived radioactive tracer $^{13}N$ is used to trace $N_2O_5$ uptake into nitrate containing aqueous aerosol particles. While uptake of $^{13}N$ labelled $N_2O_5$ to deliquesced $Na_2SO_4$ aerosol is consistent with previous studies, in presence of aerosol phase nitrate efficient exchange of labelled nitrate with the non-labelled nitrate pool was observed. The experiments provide direct evidence for efficient bulk mass accommodation of $N_2O_5$ into aqueous solution with $\alpha_b > 0.4$ at room temperature, as well as for the fast dissociation into nitronium and nitrate. While for experimental reasons this study is limited to non-acidic nitrate containing aerosol, it is likely that bulk mass accommodation and dissociation are not limiting $N_2O_5$ uptake also under wider ranges of conditions.

## 1 Introduction

The central role of $NO_x$ in the regulation of the oxidative capacity of the atmosphere is well established. $N_2O_5$, an important species of the nighttime chemistry of nitrogen oxides, has been identified as one of the major reservoir species and potential sinks of $NO_x$ (Abbatt et al., 2012; Chang et al., 2011). $NO_2$ reacts with $O_3$ to form $NO_3$, which then react together to form $N_2O_5$ (R1).

$$NO_2 + NO_3 \underset{\longleftarrow}{\overrightarrow{\hspace{2cm}}} N_2O_5$$

(R1)

Depending on location $N_2O_5$ may be removed primarily by heterogeneous hydrolysis to nitric acid or other products on aerosol surfaces, providing thus a nighttime sink for atmospheric $NO_x$ (R2) (Abbatt et al., 2012; Chang et al., 2011).

$$N_2O_5 + H_2O_{(l)} \longrightarrow 2HNO_{3(aq)}$$

(R2)

Removal of $N_2O_5$ leads to a reduction of atmospheric $NO_x$ and consequent reduction of tropospheric ozone and thus the oxidative capacity of the atmosphere (Dentener and Crutzen, 1993; Evans and Jacob, 2005). The reactivity of $N_2O_5$ with aerosols has been of ongoing interest for the past two decades. Laboratory investigations have encompassed measurement of uptake kinetics to a wide array of inorganic aerosols, like $H_2SO_4$/sulphates (Hallquist et al., 2003; Hanson and Lovejoy, 1994), NaCl/marine aerosol (Gaston and Thornton, 2016; George et al., 1994; Thornton and Abbatt, 2005), nitrate containing inorganic aerosol (Hallquist et al., 2003; Mentel et al., 1999; Wahner et al., 1998) and mineral dust (Karagulian et al., 2006; Wagner et al., 2008). Further constraints have also been derived from field observations of $N_2O_5$ and aerosol abundance and composition (Bertram et al., 2009; Phillips et al., 2016). The uptake coefficient $\gamma$ (defined as the as the probability that a gas kinetic collision of a molecule leads to its uptake at the interface) on deliquesced inorganic aerosol is in the $10^{-1}$-$10^{-2}$ range. Organic aerosol on the other hand presents a wider range of

reactivities, with values approaching those of inorganic aerosols in certain cases such as for malonic acid (Griffiths et al., 2009; Thornton et al., 2003) at around $10^{-2}$, while in other cases the uptake coefficients go down to $10^{-3}$-$10^{-5}$, such as for some polycarboxilic acids like citric acid (Gržinić et al., 2015), succinic acid or humic acid (Badger et al., 2006; Griffiths et al., 2009), long chained fatty acids and polyalcohols (Gross et al., 2009) that indicate varying effects of water content and viscosity on the reactivity and kinetic regime. Additionally, coating aqueous particles with insoluble organic films has shown to have an inhibiting effect on $N_2O_5$ uptake (Anttila et al., 2006; Folkers et al., 2003) by restricting transport from the gas phase to the aerosol phase, reduced solubility in the organic phase or limited water availability. The hydrolysis of $N_2O_5$(g) is believed to proceed through a multistep process (Behnke et al., 1997; Mozurkewich and Calvert, 1988): molecular solvation (R3) is followed by dissociation into nitronium ($NO_2^+$, or its hydrated form $H_2ONO_2^+$) and nitrate (R4) with a rate coefficient, $k_4$, estimated between $10^5$ s$^{-1}$ and $10^7$ s$^{-1}$ (Mentel et al., 1999; Bertram and Thornton, 2009; Griffiths et al., 2009). The fate of nitronium is governed by competition between the reaction with nitrate to yield molecular $N_2O_5$ again (R5) with a nearly collision limited rate coefficient, $k_5$, of $2.4 \times 10^{10}$ M$^{-1}$ s$^{-1}$ (Mentel et al., 1999), the reaction with $H_2O$(l) to yield $HNO_3$ (R7) ($k_5$, between $10^7$ and $10^9$ M$^{-1}$ s$^{-1}$ (Mentel et al., 1999; Behnke et al., 1997)) or reactions with other nucleophiles (such as chloride ion) not considered further here.

$$N_2O_{5(g)} \underset{\longleftarrow}{\longrightarrow} N_2O_{5(aq)}$$

$\qquad$ (R3)

$$N_2O_{5(aq)} \longrightarrow NO_{2(aq)}^+ + NO_{3(aq)}^-$$

$\qquad$ (R4)

$$NO_{2(aq)}^+ + NO_{3(aq)}^- \longrightarrow N_2O_{5(aq)}$$

$\qquad$ (R5)

$$NO_{3(aq)}^- + H_{(aq)}^+ \underset{\longleftarrow}{\longrightarrow} HNO_{3(aq)}$$

$\qquad$ (R6)

$$NO_{2(aq)}^+ + H_2O_{(liq)} \longrightarrow HNO_{3(aq)} + H_{(aq)}^+$$

$\qquad$ (R7)

The elementary reactions constituting this mechanism have been suggested based on the kinetic behavior as a function of a range of conditions, but have not been unambiguously been proven by detecting, e.g., $NO_2^+$ directly. As long as nitrate is a minority species in the aerosol phase, the loss of $N_2O_5$ is driven by reaction R7. Depending on the water content, either the transfer of $N_2O_5$ into the aqueous phase, R3, also referred to as mass or bulk accommodation, the dissociation (R4) or R7 are rate limiting according to current understanding (Abbatt et al., 2012).

On the other hand, in presence of significant amounts of nitrate in the aqueous phase, the loss of nitronium ion by reaction with nitrate back to $N_2O_5$, R5, becomes significant, and the uptake of $N_2O_5$ becomes lower with increasing nitrate content in the aerosol phase, typically by an order of magnitude. This effect is referred to as the 'nitrate effect' (Abbatt et al., 2012; Bertram and Thornton, 2009; Hallquist et al., 2003; Mentel et al., 1999; Wahner et al., 1998).

Most studies conducted related to $N_2O_5$ reactivity with aqueous aerosol so far were based on measuring the net gas-phase loss of $N_2O_5$ due to uptake on an aerosol. The kinetics were described with a simplified lumped mechanism by assigning an effective Henry's law constant, $H$, of $N_2O_5$ of about 2 to 5 M atm$^{-1}$ at room temperature and a net rate coefficient for the overall reaction of $N_2O_5$ with $H_2O$ to $HNO_3$, $k_0^{II}$, of around $10^5$ M$^{-1}$ s$^{-1}$ (Ammann et al., 2013; Bertram and Thornton, 2009; Griffiths et al., 2009), so that the uptake coefficient can be described via the resistor model as given by equation (1).

$$\frac{1}{\gamma} = \frac{1}{\alpha_b} + \frac{c}{4HRT\sqrt{D_l\,k^{II}[H_2O]}\,[\coth(r/l)-(l/r)]} \qquad l = \sqrt{\frac{D_l}{k^{II}[H_2O]}} \tag{1}$$

Where $\alpha_b$ denotes the bulk mass accommodation coefficient, $c$ the mean thermal velocity of $N_2O_5$ in the gas phase, $D_l$ the diffusion coefficient of $N_2O_5$ in the aqueous phase, usually set to $10^{-5}$ cm$^2$ s$^{-1}$ for low viscosity aqueous aerosol, $[H_2O]$ the concentration of $H_2O$ in M, $r$ the particle radius and $l$ the reacto-diffusive length. The apparent levelling off of $\gamma$ towards high relative humidity was explained by a limiting $\alpha_b$ of about 0.03 (Ammann et al., 2013) or by introducing a dependence of the dissociation rate, $k_4$, on the water content, leading to an apparent water content dependence of $k_0^{II}$ (Bertram and Thornton, 2009). The effect of nitrate as mentioned above has been taken into account by using equation (2) instead of a fixed rate coefficient $k_0^{II}$:

$$k^{II} = k_0^{II}\left(1 - \frac{1}{1+\dfrac{k_7[H_2O]}{k_5[NO_3^-]}}\right) \tag{2}$$

This expression requires the knowledge of the concentrations of water and nitrate and the ratio of the two rate coefficients $k_7/k_5$, for which values between 0.02 and 0.06 lead to reasonable consistency with available data (Ammann et al., 2013; Bertram and Thornton, 2009). However, the actual size of the pool of dissolved $N_2O_5$ (or its dissociated products $NO_2^+$ and $NO_3^-$), the true value of the bulk mass accommodation coefficient, $\alpha_b$, and the rate coefficients for the individual reactions remained elusive. In this study, we make specific use of the short-lived radioactive isotope $^{13}$N (Ammann, 2001; Gržinić et al., 2015; Gržinić et al., 2014) to trace the exchange of $^{13}$N-labelled nitrate resulting from uptake of labelled $N_2O_5$ with the non-labelled nitrate pool in the aerosol phase. This allows obtaining a direct estimate of $\alpha_b$ and constraints for the $N_2O_5$ dissociation (R4) and the recombination reaction (R5). The tracer method is very sensitive in detecting phase transfer of labelled $N_2O_5$ at very low gas phase concentrations and thus is able to escape any artefact nitrate effect due to build-up of nitrate during the course of a kinetic experiment for experiments with aerosol without any nitrate initially as discussed by Gržinić et al. (2015). However, this sensitivity comes at the expense of chemical selectivity in the sense that re-evaporating $HNO_3$ product from an acidic aerosol would interfere with the $N_2O_5$ measurement as shown in our previous study (Gržinić et al., 2014). We therefore performed the experiments presented here with neutral sodium sulfate and sodium nitrate aerosol to avoid this interference.

## 2. Experimental section

A detailed description of the experimental method and setup can be found in our previous publications relating to the production of $^{13}$N labeled $N_2O_5$ (Gržinić et al., 2014) and uptake of $N_2O_5$ on citric acid aerosol (Gržinić et al., 2015). Here we will present a very cursory description of the $^{13}$N production method and the changes that have been implemented to the experimental setup for the present uptake measurements.

The $^{13}$N short lived radioactive tracer (T$_{1/2} \approx$ 10 min.) is produced via the $^{16}O(p,\alpha)^{13}$N reaction in a gas target by irradiating a 20% $O_2$ in He flow with 11 MeV protons produced by the Injector II cyclotron at the Paul Scherrer Institute, Switzerland. The highly oxidized $^{13}$N labeled species are reduced to $^{13}$NO for transport via a 580 m long PVDF capillary tube to the laboratory. Figure 1 shows a schematic of the experimental setup used in this study. $^{13}$NO containing gas from the accelerator facility (50 ml/min) is mixed with ~3 ml/min of 10 ppmv non-labeled NO in $N_2$ from a certified gas cylinder and nitrogen carrier gas (only a very small fraction of NO is labeled with $^{13}$N, about $10^{-6}$ and lower). This gas flow is mixed in the $N_2O_5$ synthesis reactor with a 50 ml/min flow containing ~8 ppmv $O_3$ produced by irradiating a 10% $O_2$ in $N_2$ gas mixture with 185nm UV light. The reactor is a glass tube of 34 cm length and 4 cm inner diameter, lined with PTFE foil and operated under dry conditions to minimize hydrolysis of $N_2O_5$

on the walls. The gas phase chemistry of NO with $O_3$ results in the formation of $NO_2$, $NO_3$ and finally $N_2O_5$ (R1). The resulting gas flow is mixed with a secondary gas flow containing aerosol (720 ml/min) in an aerosol flow tube. This flow tube differs from the one that we used in our previous studies in that it is optimized for measurement of uptake coefficients in the $10^{-1}$-$10^{-2}$ range. It consists of a glass tube 39.3 cm in length and 1.4 cm in inner diameter. The inside of the glass tube is coated with halocarbon wax to minimize $N_2O_5$ losses to the wall by heterogeneous hydrolysis. A PTFE tube (6 mm diameter) is inserted coaxially into the flow tube to act as injector for the $N_2O_5$ flow. The injector contains holes at the end which allow the $N_2O_5$ flow to be injected perpendicularly to the aerosol flow. Mixing is assumed to be rapid, and a laminar flow profile is assumed to have been established in the first few cm. A black shroud is used to cover the $N_2O_5$ synthesis reactor and the aerosol flow tube in order to prevent $NO_3$ photolysis and thus $N_2O_5$ loss. Aerosol is produced by nebulizing 0.1% wt. $NaNO_3$, $Na_2SO_4$ and 1:1 $NaNO_3$/$Na_2SO_4$ solutions in MilliQ water. The resulting aerosol flow is dried over a Nafion membrane diffusion drier (RH of the drying gas outside is adjusted to 40% RH to prevent efflorescence), passed through an $^{85}$Kr bipolar ion source and an electrostatic precipitator to equilibrate the charge distribution and subsequently remove the charged particles, respectively. A humidifier is placed after the ion source to adjust the humidity of the gas flow to the required experimental levels. The overall gas flow exiting from the aerosol fast-flow reactor is split into two flows via a T-connector, one to an SMPS (Scanning Mobility Particle Sizer) system used to characterize the aerosol, consisting of a $^{85}$Kr bipolar ion source to re-establish charge equilibrium, a differential mobility analyzer (DMA, TSI 3071) and a condensation particle counter (CPC, TSI 3022). The other part of the gas flow is directed into a parallel plate diffusion denuder system, where the various $^{13}$N containing gaseous species ($N_2O_5$, $NO_3$ and $NO_2$) are trapped by lateral diffusion and chemical reaction on a series of parallel, selectively coated aluminum plates: citric acid for trapping $N_2O_5$ and $NO_3$, a 1:1 mixture of NDA (N-(1-naphtyl) ethylene diamine dihydrochloride) and KOH for trapping $NO_2$. The aerosol particles, which have small diffusivity, pass without loss through the denuder system and are trapped on a glass fiber filter placed at the exit. The radioactive decay of $^{13}$N is measured by placing a CsI scintillator crystal with PIN diode detectors (Caroll and Ramsey, USA) on each of the denuder traps and the particle filter. $^{13}$N decays by $\beta^+$ decay and the resulting positron annihilates with an electron with emission of two coincident $\gamma$-rays in opposite directions. The resulting signal can be related to the concentration of the species in the gas and particle phase by comparing the $NO_2$ concentration measured with a $NO_x$ analyzer to the $^{13}NO_2$ and $^{13}N_2O_5$ signals on the denuder plates and particle filter, respectively. The gas-phase non-labeled $NO_2$ and $N_2O_5$ concentrations were around $10^{11}$ molecule cm$^{-3}$. The signal on the particle filter is related to aerosol phase nitrate after removal of dissolved $N_2O_5$ by stripping the gas phase $N_2O_5$ in the denuder system. The equilibrium established by R4 and R5 is fast enough to ensure that. In turn, the amount of dissolved $N_2O_5$ remains orders of magnitude smaller than the amount of nitrate built up in the particle phase along the flow tube, so that the net loss of $N_2O_5$ in the gas phase corresponds to the net gain of nitrate in the particle phase. Unfortunately, due to experimental limitations, only a limited number of measurements could be performed. The $^{13}$N tracer technique is dependent on smooth operation of the accelerator facilities and constant online $^{13}$N production to be successful, which is not always the case. Individual experiments involved the acquisition of datapoints consisting of 15-20 activity measurements, each integrated over 60 s, for a given injector position. Overall, replicates from several different experiment series performed on several different operation days of the $^{13}$N delivering facility were averaged to obtain values and standard deviations reported in Table 1.

## 3. Results and Discussion

The measurements were performed at 50% and 70% RH for all three of the solutions in question. The operating procedure is similar to the one reported by Gržinić et al. (Gržinić et al., 2015; Gržinić et al., 2014), although in this case the experiment was performed by measuring the change in signals under variation of the position of the injector (reaction time) and not aerosol surface

to volume ratio. Appropriate corrections have been applied to the $N_2O_5$ signal to account for $NO_2$ interference on the denuder coating used for $N_2O_5$ (citric acid) as well as for the small amounts of $NO_3$ present in the gas phase, as described in our previous studies. Before each experimental run a wall loss measurement was performed without aerosol by moving the injector in 5-10 cm steps along the length of the aerosol flow tube and measuring the gas phase $N_2O_5$ signal, in order to obtain the pseudo-first order wall loss rate constants ($k_w$) for each experiment. These values varied between ~$3\times10^{-2}$ and ~$7\times10^{-2}$ s$^{-1}$, depending on humidity. The citric acid coating used to trap $N_2O_5$ in the denuder system is not able to differentiate $N_2O_5$ from $HNO_3$. Though, any significant presence of $HNO_3$ formed along the walls upstream of the aerosol flow tube would have been noticed due to its very large wall loss rate coefficient (Guimbaud et al., 2002), which was not observed; the uptake coefficient for labelled $N_2O_5$ on the reactor walls calculated from $k_w$ was in the range of $10^{-7}$, which would not lead to substantial $HNO_3$ production. $NO_2$ concentrations in the system were measured by switching a $NO_x$ analyzer in place of the SMPS system at the beginning of the experiment. By comparing this result to the signals from $^{13}$N labelled $N_2O_5$ and $NO_2$, the concentration of non-labelled $N_2O_5$ in the system was also calculated, as described in our previous studies (Gržinić et al., 2015; Gržinić et al., 2014). The uptake measurements were performed in a similar fashion to the wall loss measurements, although in this case aerosol was introduced at each step. By using Eq. 3, which describes gas-aerosol phase interaction kinetics, we can estimate the uptake coefficient.

$$\frac{C_p^*(t)}{C_g^*(t=0)} = \frac{1 - e^{-(k_w+k_p)t}}{1 + \dfrac{k_w}{k_p}} \tag{3}$$

$$k_p = \frac{S_p \omega \gamma_{eff}^*}{4} \tag{4}$$

where $C_g^*$(t=0) is the gas-phase labelled $N_2O_5$ concentration in molecules cm$^{-3}$ at time zero, $C_p^*$(t) is the labelled particle phase nitrate concentration expressed as number of particle phase labelled nitrate ions per volume of gas phase, in ions cm$^{-3}$, $k_w$ is the wall loss constant, measured as described above, and $k_p$ denotes the apparent first order rate constant for gas-phase loss of labelled $N_2O_5$ to the aerosol phase, which can be related to the effective uptake coefficient ($\gamma_{eff}^*$) of labelled $N_2O_5$ by Equation 4, where $S_p$ is the total aerosol surface area to gas volume ratio of the aerosol and $\omega$ is the mean thermal velocity of $N_2O_5$. Figure 2 shows the appearance of $^{13}$N in the aerosol phase as a function of time performed together with least-squares fits using Eq. (3) with $\gamma_{eff}^*$ as the only variable. The true uptake coefficients were then obtained by correction for gas phase diffusion of $N_2O_5$, as described by Pöschl et al. (2007), which leads to considerable corrections for uptake coefficients above 0.1 and particle diameters around 400 nm. The diffusion coefficient was taken as 0.085 cm$^{-2}$ s$^{-1}$ at ambient temperature and pressure (Tang et al., 2014). Table 1 shows the results obtained for the various aerosols, with corresponding experimental parameters, including $\gamma_{eff}^*$ as returned from the fits to the data and $\gamma^*$ after the diffusion correction. In absence of aerosol phase nitrate, the obtained $\gamma^*$ for $Na_2SO_4$ at 70% RH is comparable to that reported by Mentel et al. (1999). At 50% RH, it is likely that the $Na_2SO_4$ particles remained effloresced (Gao et al., 2007) (the lowest RH in the aerosol conditioning system was 40%), explaining the low $\gamma$ of 0.0053, which is comparable to that previously reported for dry ammonium sulfate (Hallquist et al., 2003). In presence of aerosol phase nitrate, uptake of $^{13}$N-labelled $N_2O_5$ is drastically higher than the net uptake of non-labelled $N_2O_5$ observed in earlier studies by (Bertram and Thornton, 2009; Griffiths et al., 2009; Hallquist et al., 2003; Mentel et al., 1999; Wahner et al., 1998) or that predicted by the IUPAC recommended expression also included in the table. A strong dependence of $\gamma^*$ on humidity is observed, with an increase by a factor about three from 50% to 70% relative humidity. On mixed $Na_2SO_4$ / $NaNO_3$ aerosol, the uptake is very close to that of pure $NaNO_3$ aerosol, at both humidities. Since the measured uptake coefficient for $^{13}$N labelled $N_2O_5$ into the nitrate containing aerosol is about an order of magnitude larger than any uptake coefficient measured into not strongly acidic aqueous aerosol in previous studies, our results

demonstrate that gas – aqueous phase exchange of $N_2O_5$ at room temperature and its dissociation are very efficient , as also suggested by Mentel et al. (1999), based on experiments with $NaNO_3$, $Na_2SO_4$ and $NaHSO_4$ aerosol.

More insight into the interpretation of the present experiments with $^{13}N$ labeled $N_2O_5$ can be obtained by deconvoluting the chemical mechanism of $N_2O_5$ hydrolysis (R3-7) to take into account the fate of the labelled and non-labelled species separately. Assuming that the labeled $N_2O_5$ molecules contain only one $^{13}N$ atom (due to the very low labeled to non-labeled NO ratio of around $10^{-6}$) and under the assumption that no isotopic effects are in play, the chemical mechanism can be assumed to proceed as shown in Figure 3a. We first assume that upon dissociation of $^{13}NNO_5$, there is an equal possibility that the $^{13}N$ label will end up either in the nitronium ion ($NO_2^+$) or in the nitrate ion ($NO_3^-$), with rate coefficients $k_4/2$. In the first scenario (right branch of the pathway for '$N^*NO_5$' in Fig. 3a) the $^{13}N$ label ends up on the nitrate ion (R4"). The corresponding non-labelled nitronium ion, if not reacting with $H_2O$, R7, reacts with the excess non-labelled nitrate to form non-labelled $N_2O_5$ (R5), while only a negligible fraction reacts with the labeled $^{13}NO_3^-$ (R5") and thus most of the labeled $^{13}NO_3^-$ is lost in the excess nitrate pool present in the aerosol. $N_2O_5$ re-evaporating to the gas phase is therefore non-labeled. Thus, in this branch, all $^{13}N$ labels remain in the aqueous phase given that evaporation as $HNO_3$ is unlikely for our neutral aerosol. As evident from the experiments, the net transfer of labelled $N_2O_5$ into the aerosol nitrate pool, and thus the combination of bulk accommodation and dissociation, is very fast.

In the other scenario (left branch of '$N^*NO_5$' pathway in Fig. 3a) of the mechanism, $^{13}N$ ends up on the nitronium ion (R4'), and there are two possibilities for its further fate. The $^{13}NO_2^+$ can react with water (R7'), bringing the label into the particulate nitrate pool. Alternatively $^{13}NO_2^+$ can react with $NO_3^-$ from the nitrate pool (R5') to reform $^{13}NNO_5$ which can eventually re-evaporate. Obviously which of these two sub-channels are prevalent depends on water and nitrate concentrations as well as the rate coefficients of the two competing reactions. The water and nitrate concentrations for the conditions of the present experiments were derived from the AIM model (Clegg et al., 1998) and are also listed in Table 1. In presence of nitrate, (R5') is the dominant pathway, as shown by Mentel et al. (1999) and Wahner et al. (1998), where values for the ratio $k_5/k_7$ of around 900 and 260, respectively, had been used as fitting parameters to explain the entirety of the nitrate effect.

In Figure 3b, we report the results of box model calculations (described in the supporting information) for the scheme shown in Figure 3a to substantiate these effects for $NaNO_3$ aerosol at 70% RH. We have used the rate coefficients for $k_4 = 5 \times 10^6$ s$^{-1}$, $k_5 = 2.4 \times 10^{10}$ M$^{-1}$ s$^{-1}$ and $k_7 = 2.6 \times 10^7$ M$^{-1}$ s$^{-1}$ as suggested by Mentel et al. (1999) for $NaNO_3$ aerosol at various humidities; the bulk accommodation coefficient $\alpha_b$ was set to one. Figure 3b (left y-axis) directly demonstrates the rapid removal of labelled nitrogen from aqueous phase $N_2O_5$ and its appearance as labelled nitrate. Parallel to that, labelled $N_2O_5$ disappears much more quickly from the gas phase and appears as aerosol phase nitrate than non labelled $N_2O_5$, as shown in Figure 3d. It is also instructive to look at the rates of individual reactions contributing to the aqueous phase budget of labelled species plotted in Figure 3b and those of the non-labelled species in Figure 3c: Even though R5' brings back half of the labelled $N_2O_5$ (rate of R5' is half of the dissociation rate, Figure 3b), the fraction of labelled $N_2O_5$ drops below half of its initial value, since the exchange reactions R4 and R5 are fast, so that it may cycle through the aqueous phase part of the scheme many times. The rate of R5" (labelled nitrate with non-labelled nitronium) is not getting competitive, so that labelled nitrate remains in the nitrate pool. The $\gamma^*$ values calculated from the net $N_2O_5$ loss rates are about 0.26 for labelled $N_2O_5$, in good agreement with the value observed in the experiment, 0.29, and 0.0033 for non-labelled $N_2O_5$. The latter is consistent with the observation by Mentel et al. but lower than that suggested by the parameterization given in equation (1) based on the IUPAC recommendation and listed in Table 1 or that suggested by Bertram and Thornton (2009) or Griffiths et al. (2009) developed based on a wider data set with other aerosol compositions. Box model simulation of the mixed salt case (not shown), also at 70% RH, returns essentially the same results, 0.25 and 0.009, for the uptake coefficient of labelled and non-labelled $N_2O_5$, respectively, also in line with our measured $\gamma^*$ of 0.26 for labelled $N_2O_5$. The right column of Figure 3, plots e,f,g,h, present the box model results for $Na_2SO_4$ aerosol at 70% RH. The main difference is that in this case, R5 represents

only a negligible pathway for non-labelled nitronium resulting from R4'', so that the majority of labelled nitrate recycles back to labelled $N_2O_5$ via R5''. As a result, both labelled and non-labelled species behave exactly parallel, both in terms of rates and concentration. When solubility and rate coefficient are taken the same as those for the nitrate aerosol, the magnitude of the calculated uptake coefficient is an order of magnitude too high. The difference between the kinetics described by Mentel et al. (1999) and Bertram and Thornton (2009) is that overall uptake is limited by the dissociation, $k_4$, in the latter, while Mentel et al. suggest faster dissociation and limitation by the interplay of the reactions of $NO_2^+$ with $H_2O$ and $NO_3^-$. Replacing $k_4$ by the parameterization suggested by Bertram and Thornton, keeping $k_5$ as before, but replacing $k_7$ such that $k_7/k_5$=0.06 as suggested by Bertram and Thornton, yields $\gamma = 0.06$, in reasonable agreement with the uptake coefficient measured here (0.027). However, this can as well be achieved by the Mentel et al. kinetics scheme via lowering $k_7$ by one order of magnitude to achieve lower net uptake, which is what has been used in the calculations shown in Figure 3f,g,h. In view of the limited dataset it has been beyond the scope of this study to retrieve more precise rate coefficients for R4, R5 and R7. The present results clearly indicate that accommodation and dissociation of $N_2O_5$ are not limiting net uptake for aerosol containing substantial amounts of nitrate. Furthermore, these experiments provide direct evidence that these elementary steps actually exist, because otherwise exchange with the non-labelled nitrate pool would not occur.

The measured uptake coefficient increases with RH (water content) as the increase in water concentration promotes the reaction of $^{13}NO_2^+$ with water (R7') while at the same time a decrease in nitrate concentration suppresses the $^{13}NNO_5$ reformation sub-channel (R5'). However, as argued above, in presence of nitrate at several M, the dominant fate of the label in the left branch of the mechanism in Fig. 3a is the reformation of labeled $N_2O_5$. If bulk accommodation would limit uptake, the measured uptake coefficient would be only slightly larger than $\alpha_b/2$, with $\alpha_b/2$ contributed by the right branch of the mechanism, and a minor contribution from the left branch. Our results therefore strongly indicate that $\alpha_b$ must be at least about twice as high as the uptake coefficient of $^{13}N$-labelled $N_2O_5$ measured for $NaNO_3$ and $NaNO_3/Na_2SO_4$ aerosols and thus $\alpha_b$>0.4 when including the uncertainties related to aerosol surface to volume ratio and correction for gas phase diffusion. In the box model simulations discussed above, $\alpha_b$ has been set to one. The only systematic uncertainty not considered in this is the assumption that isotope effects do not influence the symmetry of the disproportionation reaction (R4).

Interestingly, the measured uptake coefficient is more sensitive to the water activity, irrespective of whether it is pure nitrate or mixed nitrate / sulfate particles, than to the fractional nitrate content. In spite of the fact that the expected ratio $k_7[H_2O]/k_5[NO_3^-]$ (see Table 1) increases substantially from pure nitrate to mixed nitrate / sulfate, still bulk accommodation and dissociation are not limiting, and the efficient exchange of the label with the non-labelled nitrate pool dominates the behavior. Thus an increase of $\alpha_b$, solubility in terms of salting effects, or of the dissociation could explain the increase of the uptake coefficient of labelled $N_2O_5$ by a factor of four from 50 to 70 % RH.. As discussed first by Mentel et al. (1999), and later by Griffiths et al. (2009) and Bertram and Thornton (2009), the kinetic regime may shift between limitation by R7, R4 or R3 between low and high water activity, and it remains difficult to unambiguously assign the kinetic regime. Also from the present study, the evidence for the fast dissociation is very strong for the high nitrate mole fractions used, and we can only suggest that the dissociation is similarly fast for the salt aerosol in absence of nitrate.

On a side note, in principle, we cannot exclude a surface contribution. Since also the suggested bulk accommodation is followed via obviously very fast dissociation into nitronium and nitrate, this process will occur close to the surface anyhow, with a short reacto-diffusive length (in contrast to the much slower hydrolysis reaction of nitronium with $H_2O$). Therefore, for instance, varying the surface to volume ratio of the aerosol would not allow differentiating bulk from surface accommodation. Though, we would prefer not invoking a surface process if bulk phase processing explains the observations.

Obviously, the fast isotope exchange that allows tracking the dissociation of dissolved $N_2O_5$ depends on the presence of a nitrate pool, which has to be substantial enough to 'trap' the [13]N tracer and prevent its release. In other words, this method traces uptake of $N_2O_5$ into the non-limiting pool of aqueous nitrate. In absence of nitrate, R7' and R7 dominate in both branches (Fig. 3e), and the measured uptake coefficient for the [13]N labelled $N_2O_5$ is comparable to that reported with other techniques for the same aerosol, in the range of a few $10^{-2}$ for $Na_2SO_4$ at 70% RH in this study or for $(NH_4)_2SO_4$ reported by Gržinić et al. (2014), both consistent with available literature. For comparison, calculated uptake coefficients based on the parameterization and values from the IUPAC evaluation (Ammann et al., 2013), implemented in equation (1) and (2), are also listed in Table 1.

**Conclusion and atmospheric implications**

Highly efficient uptake of [13]N-labelled $N_2O_5$ into nitrate containing aqueous aerosol was observed and attributed to the exchange of [13]N labeled nitrate as dissociation product of $N_2O_5$ with the nitrate pool in the aqueous phase. This allows suggesting a very large value for the bulk mass accommodation coefficient for $N_2O_5$ into an aqueous aerosol at room temperature of $\alpha_b > 0.4$ at high relative humidity and fast dissociation at $> 10^6$ s$^{-1}$. This also provides direct evidence that the fast dissociation into nitrate and nitronium at $> 10^6$ s$^{-1}$ actually occurs, and thus also supports the arguments behind the nitrate effect. The observed behavior of [13]N-labelled $N_2O_5$ is similar to that observed by Wachsmuth et al. (2002), where [83-86]Br isotopes have been used to determine the bulk accommodation coefficient of HOBr on aqueous bromide containing aerosol. The large value of $\alpha_b$ obtained for $N_2O_5$ at room temperature implies that other limiting processes must be at work to explain the insensitivity of the uptake coefficient to water content at high relative humidity (Abbatt et al., 2012; Griffiths et al., 2009; Thornton et al., 2003). Possible other aspects may be the temperature dependence of the solubility of $N_2O_5$ or salting effects as discussed in these previous studies, for which also strong indications come from this study to explain the strong water activity dependence of the observed uptake coefficient of [13]N labelled $N_2O_5$. At least this study would exclude any accommodation limitation and supports previous indications that reaction and diffusion limit uptake of $N_2O_5$ to low viscosity aqueous aerosol (Gaston and Thornton, 2016).

**Acknowledgements**

The authors would like to thank the staff of the PSI accelerator facilities and of the isotope production facility IP-2 for their invaluable help during experimental work. Technical support by M. Birrer is much appreciated. This study was supported by the Swiss National Science Foundation (grants no. 130175 and 149492).

**The Supplement related to this article is available online**

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

**Table 1.** Experimental parameters and results

| Aerosol | $NaNO_3$ | $NaNO_3$ | $NaNO_3/$ $Na_2SO_4$ | $NaNO_3/$ $Na_2SO_4$ | $Na_2SO_4$ | $Na_2SO_4$ |
|---|---|---|---|---|---|---|
| Molar mixing ratio | - | - | 1:1 | 1:1 | - | - |
| Temperature (K) | 295±1 | 295±1 | 295±1 | 295±1 | 295±1 | 295±1 |
| RH (%) | 50±1 | 70±1 | 51±1 | 70±1 | 51±1 | 70±1 |
| Average S/V ratio ($m^2/m^3$) | $5.66\times10^{-3}$ | $5.00\times10^{-3}$ | $3.36\times10^{-3}$ | $4.76\times10^{-3}$ | $3.42\times10^{-3}$ | $3.00\times10^{-3}$ |
| $\gamma^*_{eff}$ ($^{13}NNO_5$) | 0.057 | 0.18 | 0.054 | 0.17 | 0.0052 | 0.025 |
| Error[a] | ±0.004 | ±0.02 | ±0.008 | ±0.02 | ±0.002 | ±0.003 |
| $\gamma^*$ ($^{13}\mathbf{NNO_5}$) [b] | **0.067** | **0.29** | **0.063** | **0.26** | **0.0053** | **0.027** |
| Total error [c] | ±0.02 | ±0.1 | ±0.02 | ±0.2 | ±0.002 | ±0.008 |
| $[H_2O]$ (M) [d] | 28.5 | 38.2 | 34.4 | 41.6 | 38.6 | 43.8 |
| $[NO_3^-]$ (M) [d] | 13.1 | 8.78 | 4.60 | 3.29 | 0 | 0 |
| $k_7[H_2O]/k_5[NO_3^-]$ [e] | 0.044 | 0.087 | 0.15 | 0.25 | - | - |
| $\gamma$ (non-labelled $N_2O_5$) [f] | 0.0050 | 0.010 | 0.013 | 0.019 | 0.044 | 0.047 |

[a] 95% confidence level of the fit of eq. (3) to the data as shown in Fig. 2

[b] corrected for diffusion in the gas phase

[c] including 30% uncertainty related to the aerosol surface to volume ratio

[d] from AIM model (Clegg et al., 1998)

[e] $k_7/k_5 = 0.02$

[f] calculated using equations (1) and (2), with $k_0^{II} = 10^5$ $M^{-1}$ $s^{-1}$ and $H = 2$ M atm$^{-1}$

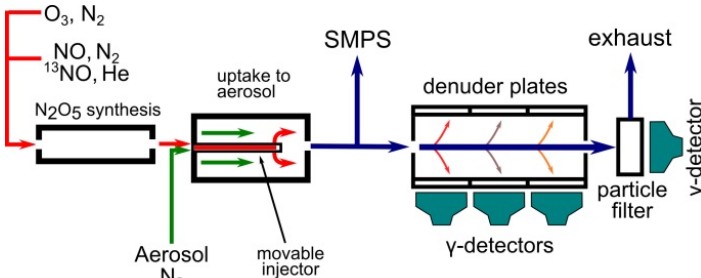

**Fig. 1.** Schematic of the modified experimental setup used in this study

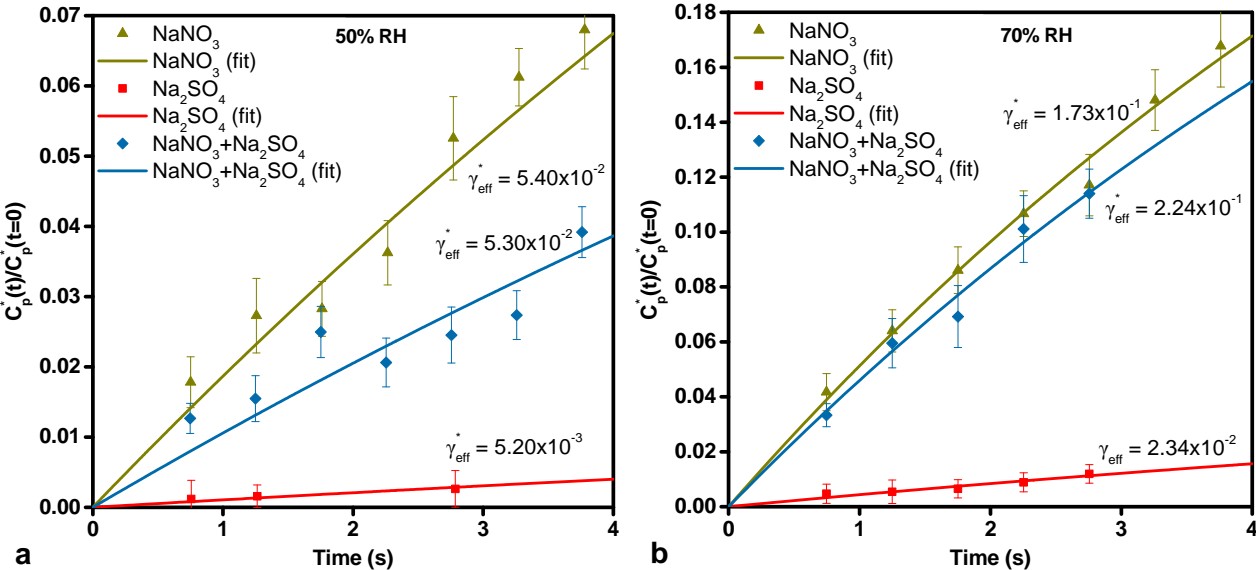

**Fig. 2.** Normalized particle-phase labelled nitrate concentration vs. time graph for experiments performed at 50% RH (a) and 70% RH (b). The data points represent the measurement data, the error bars represent the standard deviation of the measurements, the curves are a least-squares fit of Eq. 1 to the measured data.

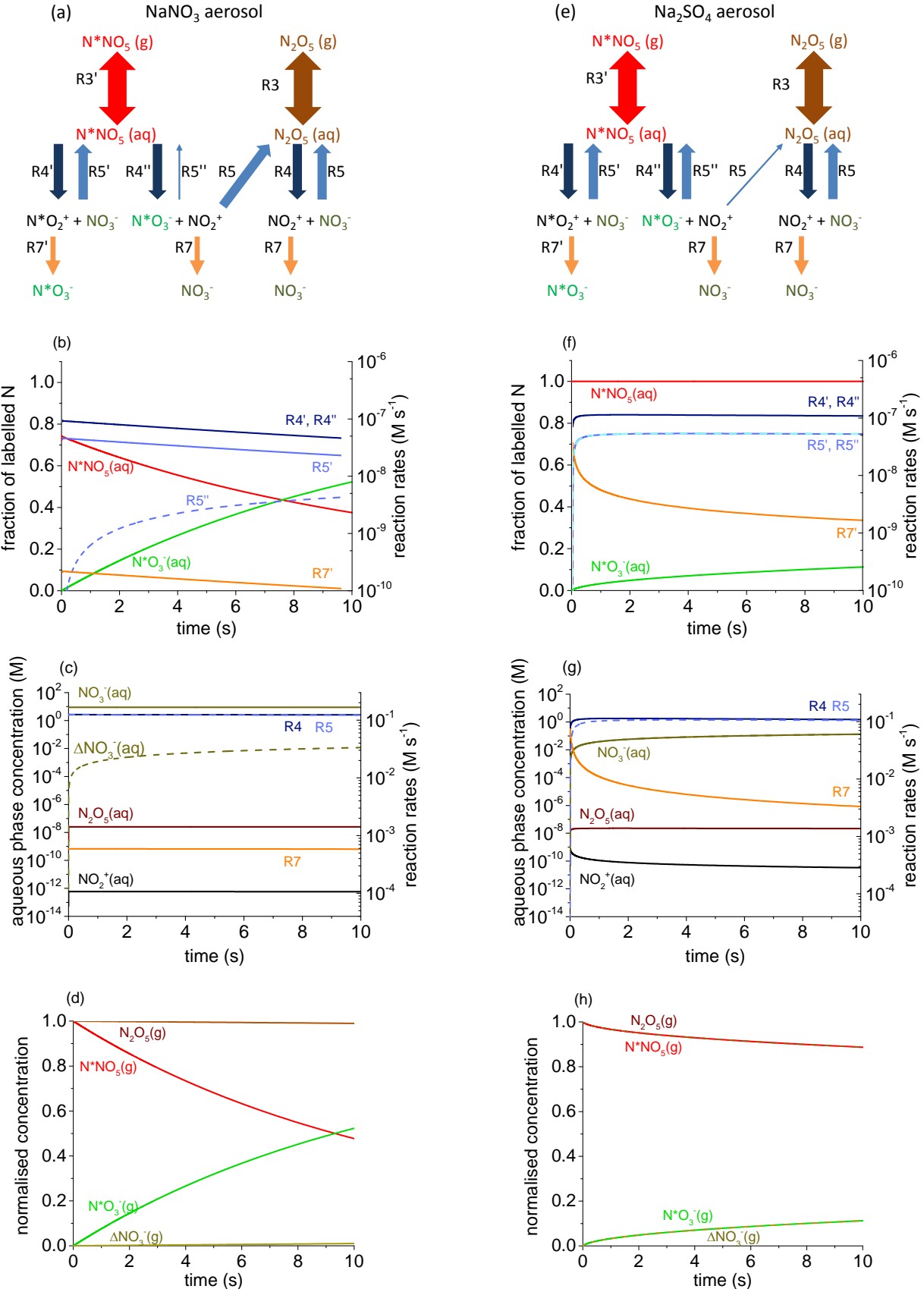

**Fig. 3.** (a) The chemical mechanism of $N_2O_5$ hydrolysis and chemical isotope exchange in nitrate containing solutions. N* refers to $^{13}N$. (b, c, d) Results of box model calculations for $NaNO_3$ aerosol at 70% RH: evolution as a function of time of: (b) the fraction of labelled $N_2O_5$ in the aqueous phase (red, left axis) and ratio of labelled nitrate to the total number of labeled $N_2O_5$ initially available (green, left axis); rates of reactions relevant for the budget of aqueous phase $^{13}N$ (light to dark blue, right axis); (c) rates of reactions relevant for the budget of non-labelled species in the aqueous phase; (d) evolution of normalized gas phase $N_2O_5$ (labelled, red; non-labelled, brown) and aqueous phase nitrate (labelled, green; non-labelled, olive) as a function of time. (e, f, g, h) scheme and plots of the same quantities, but for $Na_2SO_4$ aerosol at 70% RH in absence of pre-existing nitrate.