# Peer review of "Efficient bulk mass accommodation and dissociation of N2O5 in neutral aqueous aerosol"

_Atmospheric Chemistry and Physics, 2016_

## Referee Comment (RC1) · Anonymous Referee #1 · 31 Dec 2016

This manuscript describes observations of the loss of radioactively labelled N2O5 upon exposure of this gas to neutral sulfate and nitrate aerosol. The hydrolysis reaction of N2O5 is one of the most important tropospheric heterogeneous reactions, leading to significant rates of NOx loss. Although the kinetics of this process have been studied for a long time there still remain interesting and important outstanding questions concerning its mechanism. This paper rightly addresses one of those issues, in particular the mass accommodation coefficient of N2O5 on aqueous particles. For context, in past studies it has been observed that reactive uptake coefficients tend to saturate at values somewhat larger than 0.01, suggesting that a mass accommodation coefficient of this size may be the limiting factor.

The authors use a unique experimental approach whereby they use short lived radioactively labelled N (N*) which is incorporated into N2O5, which is then injected into an

aerosol kinetics flow tube. The experimental techniques are all well-tested, appropriate and well-described. The group has a lot of expertise in this area. The key, important aspect of this technique is that ultra trace levels of N* can be measured in the aerosol particles, providing additional observational data that most other experiments (for example those that monitor the gas phase alone) do not.

The main result from this paper is that the observed uptake coefficient for labelled N2O5 onto nitrate particles is much larger than 0.1, which the authors indicate is a measure of the bulk mass accommodation coefficient into aqueous particles being this large in general.

I have a few questions:

1. I am a bit confused by the nomenclature. In Figure 2 (and Equation 3) there is a [N2O5(p)] term. What is meant by that? It is clearly not the dissolved concentration of N2O5 in the particle. Is it instead the concentration of N* in the particle (as stated in line 37)? But this would be orders of magnitude lower than the level of dissolved unlabeled N. So, how do the authors scale N* concentrations to indicate the amount of N2O5 that has been taken up to the particle, which is what I believe this term represents? Also please use either C or [ ] nomenclature for concentration but not both.

2. I must be missing a very important point in the paper, and would like the authors to explain this to make the paper clearer. In particular, I understand that the kinetics experiments with nitrate particles indicate a large uptake coefficient that can be interpreted as a large mass accommodation coefficient. But, I am at a loss to understand why the results with sulfate particles, that show roughly an order of magnitude smaller uptake, cannot also be interpreted as a mass accommodation coefficient measurement. If the reaction proceeds half through the left channel in Figure 3, then the labelled nitrate so formed should stay in solution. If the mass accommodation coefficient is really larger than 0.1 on aqueous droplets then the uptake coefficient in these experiments should have been at least this large. But the observations show much

smaller uptake kinetics. Does this not imply that the mass accommodation coefficient on sulfate must not be as large as on nitrate?

3. The results are interpreted in terms of bulk processes. With very high concentrations of nitrate, could the chemistry for those particles be happening on the particle surface? In this case the kinetics would be surface-uptake controlled and not bulk-controlled, and then this would not be a measurement of the bulk mass accommodation coefficient. Overall, is it possible that the observations are consistent with (fast) surface uptake for the nitrate experiments and with (slower) bulk uptake for the sulfate experiments?

4. It is stated a few times that nitric acid does not volatilize under the conditions of the experiments (and this was the reason to use neutral particles and not acidic ones). Can the authors confirm that this is the case for their conditions?

5. Table 1: Please list how many experiments were conducted per experimental condition.

---

## Referee Comment (RC2) · Anonymous Referee #2 · 2 Jan 2017

Review of Grzinic et al., Efficient bulk mass accommodation of N2O5 into neutral aqueous aerosol

Summary and General Comments: Grzinic et al presents a timely analysis of the reactive uptake of 13N labelled N2O5 to neutral aqueous nitrate and sulfate aerosol. The analysis permits assessment of N2O5 mass accommodation and the validity of the N2O5 disproportionation mechanism. The authors find high values for N2O5 mass accommodation (> 0.4) and the best evidence to date for the concerted N2O5 ionization mechanism, involving the nitronium ion intermediate. The results are a welcome addition to the field and should be published following the authors attention to the following general comments:

1) Most of the discussion centers on the uptake of N2O5 to nitrate aerosol. The con

clusions drawn suggest large values for alpha and supports N2O5 ionization at the particle interface. What is less clear is if this is a general result for all aerosol at the same water activity. In the case of sulfate particles, labelled N2O5 likely evaporates from the interfacial region wo/the opportunity for exchange with unlabeled nitrate. The upper limit of 0.03 is achieved that is likely a combination of alpha, KH, and the reaction rates described. If KH and the hydrolysis rates are the same for sulfate and nitrate aerosol, is it correct to generalize the mass accommodation results derived from the nitrate aerosol experiments to all aqueous aerosol of comparable interfacial water activity? It would be helpful for the authors to provide some discussion on these points and the generality of the derived mass accommodation coefficient. I would find it very helpful if there was a second panel to Figure 3, which showed the processes for the sulfate particles.

2) Most of the aerosol flow reactor community is familiar with the kinetic equations shown, for unlabeled reactants. Are there any special considerations that need to be accounted for regarding single, vs multiple collisions of the labelled N2O5 with aerosol and the walls? Or rather, in Fig. 3, how many times would you expect a labelled N2O5 to cycle through a particle or the wall of the flow tube over the time constant of the flow reactor? Is this important to the analysis or derivation of the equations presented?

3) What is the effect of labelled HNO3 that is generated in the source region? How would this be interpreted in the experiment. It is not uncommon for N2O5 sources to be 10:1 HNO3 to N2O5. What is the expected ratio in this experiment?

4) Like reviewer #1, I am also confused by the notation of [N2O5]p. It would be helpful if equation 3 and Figure 2 were consistent in notation. It would also be helpful to denote between labelled and unlabeled here, as the unlabeled N2O5 uptake coefficient to nitrate aerosol could be pretty small even at high labeled N2O5 uptake coefficients, correct?

---

## Referee Comment (RC3) · Anonymous Referee #3 · 3 Jan 2017

This paper describes a novel experiment to probe the mass accommodation coefficient of N2O5 on deliquesced sodium sulfate and sodium nitrate aerosol particles. The mass accommodation coefficient is a fundamental parameter needed to describe and understand the reactive uptake of N2O5 and its conversion to products such as nitrate and nitryl halides on atmospheric aerosol particles. The experiment relies on using radioactive 13N2O5, and pH neutral (aqueous) aerosol particles, followed by collection of the particles and measurement of the 13N nitrate products. Overall this is an interesting experiment, and I recommend publication if some fairly significant issues can be clarified or addressed.

If I understand correctly, the net uptake is determined by measuring the gas-phase 13N and the particle phase 13N, where it is assumed particle phase 13N was only derived from 13N N2O5, and that any 13N nitrate produced from N2O5 stays in the particle

phase.

Lines 25 on – what are the relative amounts of NO2 and N2O5. NO2 is known to undergo slow disproportionation to HNO3. Is this a concern here? What about wall production of 13HNO3 and its incorporation into the particles due to revolatilization?

I'm not sure that the net reaction probabilities and their interpretation is correct or at least well explained. I can see why the net uptake measured in this 13N system on nitrate containing particles would be higher than that for unlabeled N2O5 on nitrate containing particles, due to the added channel of "isotope exchange" (for short). But I do not see why there would be an order of magnitude larger net reactive uptake of 13N N2O5 on nitrate containing particles compared to 13N N2O5 reacting on sulfate particles. I think a simple box model using the ratio of k5/k7 and nitrate and water molalities might help.

Or the description of the results and their interpretation needs to be clearer. I don't see where the limitation in the sulfate system is to produce that much lower of a net reaction probability, if there is equal probability of 13N N2O5 becoming 13NO3- and 13NO2+. What would be the limitation to 13NO2+ in the sulfate particles that doesn't exist in the nitrate particles?

Lines 10 onward, page 2 – I don't think all previous estimates of bulk accommodation were as low as implied here, only that net reactive uptake was lower. In other papers, a typical assumption was also that the accommodation coefficient was not limiting net uptake, see for e.g. Bertram and Thornton ACP 2009 and others.

Line 23, page 5 – sentence that starts here as a double negative at the end, making it difficult to interpret the meaning. Suggest: Thus, in this branch, all 13N labels remain in the aqueous phase given that HNO3 evaporation from neutral aerosol is unlikely.

Line 7, page 6 – this paragraph is interesting. In the Bertram and Thornton parameterization, there is a dependence of the equilibrium between N2O5(aq) = NO3-(aq)

+ NO2+(aq) on the water activity (molar concentration in that paper) that was inferred from the behavior of observed uptake coefficients. This dependence was in addition to that inferred for the fate of the NO2+. Seems this presents a possible physical explanation.

Conclusions, last sentence: "The absence of accommodation limitation also helps rationalizing results from field experiments (Phillips et al., 2016; Wagner et al., 2013), where under some conditions uptake coefficients constrained by combined N2O5 and aerosol measurements are larger than those suggested by laboratory studies available to date."

I'm not sure I completely agree with this conclusion. Do the measurements presented herein actually help explain field determinations of the reaction probability being larger than those measured in the laboratory? A few lines above, it was noted that the net reaction probability from this set of measurements was similar to previous values (at least for sulfate particles). The field observations are not deriving accommodation coefficients, but net reaction probabilities. I agree that if some previous laboratory studies concluded mass accommodation coefficients were <0.04, then field observations of net reaction probability greater than 0.04 would not be supported by such a conclusion (or vice versa).

However, other laboratory studies and even parameterizations for similar aerosol systems as those used here, have concluded or assumed mass accommodation is not limiting (e.g. Bertram and Thornton, ACP 2009 and references therein), and thus that the limitations are in the chemistry and diffusion.

Thus, I would argue that for the reaction probabilities derived from field measurements that are larger than some laboratory measurements (pretty small fraction) are because of three possible reasons: (1) laboratory measurements have an additional limitation to net uptake not present (or reduced) in the atmosphere (could be salting out as suggested), (2) there are additional faster reaction pathways for N2O5 in ambient aerosol

particles not yet probed in the laboratory, or (3) because the assumptions required in deriving the reaction probabilities from field observations are flawed.

It is hard to imagine there is something that reacts with NO2+ much faster than Chloride in solution (though it is possible), and in aqueous sea salt particles, the reaction probability is still only measured to be similar to that on aqueous sulfate particles. That to me suggest the limitation that seems to keep reaction probabilities < 0.05 at room temperature is less likely the NO2+ chemistry and instead the solubility, dissociation, and diffusion beforehand. Anyway, that is a long winded way of saying that I don't think there is support for that conclusion as written.

---

## Referee Comment (RC4) · Anonymous Referee #4 · 6 Jan 2017

General comments

The authors present results describing an important atmospheric chemical process, the removal of the NOx reservoir by aerosol, and the article contains new data which contribute to the development of physico-chemical models of the uptake process. The experiments are novel in their use of an isotopically labelled N2O5.

Uptake of N2O5 is quantified by the appearance of 13N in the aerosol phase, as determined by measurements of aerosol deposited onto a filter, and this quantity is used to infer the rate of uptake via fits of the ratio of N2O5 in the particle phase to the initial gas phase N2O5 concentration (measured as 13N deposited onto a citric acid denuder) as a function of gas-aerosol contact time. The authors derive gamma(eff) from these fits, which are then corrected for gas-phase diffusion effects, to give true uptake coeffi-

cients, gamma.

The systems studied are sodium nitrate and sodium sulfate aerosol, and aerosols of mixed composition, at 50% and 70% RH and room temperature.

The most surprising result is in the case of the nitrate aerosol. The uptake coefficient for the nitrate aerosol in the range 50%-70% RH has been previously determined to be small, indicating a slow net rate of loss of N2O5 from the gas phase into the aerosol. The authors find the uptake coefficient as determined by the appearance of 13N in the aerosol phase indicates efficient accommodation and reaction with bulk aerosol nitrate. More surprising still, the rate of uptake is larger than previously reported accommodation coefficients, which was derived from the observation that uptake coefficients tend to a maximum value of around 0.04 under all conditions. Thus, the accommodation of N2O5 would appear to be much more efficient than previously considered. This is an interesting result.

The experimental method is well-established and the group have considerable expertise in this area. The paper is generally well-written, figures are clear, and appropriately referenced. While the number of systems studied is rather small, and the paper is consequently rather brief, the difficulty of the measurements and their interest merits publication at this stage. I do feel the paper might be improved with some more quantitative treatment of the kinetics of the uptake process, but would be happy to see the work published at this stage following minor revisions.

Specific Comments

I'd like the authors to include an estimate of the mixing ratio of N2O5 in the flow tube, as well as HNO3, as well as Va and Sa values (needed in the diffusion correction and to establish the role of possible aerosol size effects in the uptake process).

The results hang together well but the low uptake onto sodium sulfate aerosol needs to be discussed in more detail. In particular, the question needs to be addressed as

to why is it so low - lower than the reported value by Wahner et al. of 0.037 (+ 0.008 - 0.019)?

The aerosol is dried to 40% RH ('to prevent effloresence' line 39, page 2). This is quite far below the reported efflorescence RH of Na2SO4 of 58%, (e.g. quoted in J. Phys. Chem. A, 2007, 111 (42), pp 10660–10666), so partial or complete efflorescence may explain the low values of uptake observed for the 50% RH case of Na2SO4, as uptake onto solid sulfate is known to be slower. Could this be a possible reason?

I think the discussion of salting effects requires more detail (do they have any evidence for the salting they propose?) and to be better linked to variation in RH.

––––––––––––––––––––––––––––

---

## Author Comment (AC1) · 16 Mar 2017

The authors would like to express their appreciation for the interest, lively discussion and constructive comments of our manuscript. This has led to some detailed analysis by box model calculations that will be included in the revised version. A detailed response to all reviewer comments is given in response to each individual review.
* * *

---

## Author Response (AR1)

**Replies to comments for article "Efficient bulk mass accommodation of N2O5 into neutral aqueous aerosol" by G. Gržinić et al.**

Goran Gržinić [1,2], Thorsten Bartels-Rausch[1], Andreas Türler[2,3], Markus Ammann[2]

[1] Laboratory of Environmental Chemistry, Paul Scherrer Institut, Villigen, 5232, Switzerland
[2] Department of Chemistry and Biochemistry, University of Bern, Bern, 3012, Switzerland
[3] Laboratory of Radiochemistry, Paul Scherrer Institut, Villigen, 5232, Switzerland

*Correspondence to*: Markus Ammann (markus.ammann@psi.ch)

**Anonymous Referee #1**

**1. I am a bit confused by the nomenclature. In Figure 2 (and Equation 3) there is a [N2O5(p)] term. What is meant by that? It is clearly not the dissolved concentration of N2O5 in the particle. Is it instead the concentration of N\* in the particle (as stated in line 37)? But this would be orders of magnitude lower than the level of dissolved unlabeled N. So, how do the authors scale N\* concentrations to indicate the amount of N2O5 that has been taken up to the particle, which is what I believe this term represents? Also please use either C or [ ] nomenclature for concentration but not both.**

*Response:* Thank you for pointing this out. Since the observables of the present experiment are gas phase labelled N₂O₅ and labelled product, thus labelled nitrate, in the aerosol phase, Equation (3) should more clearly refer to the fate of labelled molecules. We will choose a more clear notation and explain it more explicitly in the text to avoid confusion.

*Action in manuscript:* notation in equation (3) and Figure 2 changed

**2. I must be missing a very important point in the paper, and would like the authors to explain this to make the paper clearer. In particular, I understand that the kinetics experiments with nitrate particles indicate a large uptake coefficient that can be interpreted as a large mass accommodation coefficient. But, I am at a loss to understand why the results with sulfate particles, that show roughly an order of magnitude smaller uptake, cannot also be interpreted as a mass accommodation coefficient measurement. If the reaction proceeds half through the left channel in Figure 3, then the labelled nitrate so formed should stay in solution. If the mass accommodation coefficient is really larger than 0.1 on aqueous droplets then the uptake coefficient in these experiments should have been at least this large. But the observations show much smaller uptake kinetics. Does this not imply that the mass accommodation coefficient on sulfate must not be as large as on nitrate?**

*Response:* This is indeed an important point of the manuscript, and obviously we need to clarify this better. In absence of aerosol phase nitrate, labelled nitronium not undergoing hydrolysis will react with non-labelled nitrate coming into the aqueous phase as well and reform N₂O₅ and equilibrate with the gas phase again. Therefore, in absence of nitrate, uptake of labelled N₂O₅ is exactly following the fate of non-labelled N₂O₅. The fast accommodation and dissociation of labelled and non-labelled N₂O₅ into the aqueous phase is not becoming rate limiting. The amount of aqueous phase N₂O₅, nitronium and nitrate is in effective

Henry's law equilibrium with gas phase $N_2O_5$, nitrate from the reaction of nitronium with water is only slowly building up. We will show some box model simulations to show this behavior in the revised version.

*Action in manuscript:* notation in equation (3) and Figure 2 changed

**3. The results are interpreted in terms of bulk processes. With very high concentrations of nitrate, could the chemistry for those particles be happening on the particle surface? In this case the kinetics would be surface-uptake controlled and not bulk-controlled, and then this would not be a measurement of the bulk mass accommodation coefficient. Overall, is it possible that the observations are consistent with (fast) surface uptake for the nitrate experiments and with (slower) bulk uptake for the sulfate experiments?**

*Response:* In principle, we cannot exclude a surface contribution. Since also the suggested bulk accommodation is followed via obviously very fast dissociation into nitronium and nitrate, this process will occur close to the surface anyhow, with a short reacto-diffusive length (in contrast to the much slower hydrolysis reaction of nitronium with $H_2O$). Therefore, for instance, varying the surface to volume ratio of the aerosol would not allow differentiating bulk from surface accommodation. Though, we would prefer not invoking a surface process if bulk phase processing explains the observation. But we will place a caveat on this in the revised version.

*Action in manuscript:* short paragraph added towards the end of the results and discussion section.

**4. It is stated a few times that nitric acid does not volatilize under the conditions of the experiments (and this was the reason to use neutral particles and not acidic ones). Can the authors confirm that this is the case for their conditions?**

*Response:* it is correct that progressing uptake of non-labelled $N_2O_5$ during the residence time in the flow tube leads to acidification of the particles. However, with non-labelled N2O5 being in the low ppb range, the maximum $HNO_3$ concentration in the aqueous phase becomes about $10^{-3}$ M leading to a pH of around 3 so that evaporation should not play a role. As explained in Grzinic et al. (2014), measurable evaporation could be observed in the denuder detection system on the traps following that for $N_2O_5$ but was not observed in the present study. Though, unambiguous differentiation of that signal between $HNO_3$ evaporating from the particles and $N_2O_5$ formed from the back reaction of nitrate with nitronium would not be possible.

*Action in manuscript:* effects of $HNO_3$ described at the beginning of the results and discussion section.

**5. Table 1: Please list how many experiments were conducted per experimental condition.**

*Response:* Each experiment involved 15-20 activity measurements, each averaged over 60 s, for each injector position. Overall, replicates from several different experiment series

performed on several different operation days of the $^{13}$N delivering facilities were averaged to obtain values and standard deviations reported in the table. This is due to the often only limited periods of stable $^{13}$N production. This will be added to the experimental section.

*Action in manuscript:* information added to the end of the experimental section.

**Replies to comments for article "Efficient bulk mass accommodation of N2O5 into neutral aqueous aerosol" by G. Gržinić et al.**

Goran Gržinić [1,2], Thorsten Bartels-Rausch[1], Andreas Türler[2,3], Markus Ammann[2]

[1] Laboratory of Environmental Chemistry, Paul Scherrer Institut, Villigen, 5232, Switzerland
[2] Department of Chemistry and Biochemistry, University of Bern, Bern, 3012, Switzerland
[3] Laboratory of Radiochemistry, Paul Scherrer Institut, Villigen, 5232, Switzerland

*Correspondence to*: Markus Ammann (markus.ammann@psi.ch)

**Anonymous Referee #2**

**1) Most of the discussion centers on the uptake of N2O5 to nitrate aerosol. The conclusions drawn suggest large values for alpha and supports N2O5 ionization at the particle interface. What is less clear is if this is a general result for all aerosol at the same water activity. In the case of sulfate particles, labelled N2O5 likely evaporates from the interfacial region wo/the opportunity for exchange with unlabeled nitrate. The upper limit of 0.03 is achieved that is likely a combination of alpha, KH, and the reaction rates described. If KH and the hydrolysis rates are the same for sulfate and nitrate aerosol, is it correct to generalize the mass accommodation results derived from the nitrate aerosol experiments to all aqueous aerosol of comparable interfacial water activity? It would be helpful for the authors to provide some discussion on these points and the generality of the derived mass accommodation coefficient. I would find it very helpful if there was a second panel to Figure 3, which showed the processes for the sulfate particles.**

*Response:* we agree that the present results are strictly valid only for the compositions used in the present experiments. Since the observed gamma values for the pure sodium sulfate case follow that expected for non-labelled $N_2O_5$, labelled $N_2O_5$ seems to follow the fate of non-labelled $N_2O_5$, since not enough non-labelled nitrate is building up to allow exchange with that pool and larger uptake for labelled $N_2O_5$. Therefore, in fact, strictly speaking we are unable to prove that alpha and dissociation in aqueous aerosol devoid of nitrate is as fast as in presence of nitrate. Nevertheless, it is reasonable to assume (and all other laboratory studies support this) that water activity drives the reactivity with $N_2O_5$ apart from the nitrate effect and the particle physical properties. Instead of a second panel to Figure 3, we will provide some results of box model simulations to demonstrate the different behavior of labelled and non-labelled $N_2O_5$ for the two cases. Unfortunately, the dataset is not large enough in terms of nitrate concentration range to more precisely constrain the individual rate coefficients as a function of water activity.

*Action in manuscript:* Apart from providing more details by using the box model calculations of both nitrate and non-nitrate containing cases, the discussion is now clearly differentiating between conclusions for nitrate containing aerosol as derived from the present experiments, and possible implications for aqueous aerosol of different composition.

**2) Most of the aerosol flow reactor community is familiar with the kinetic equations shown, for unlabeled reactants.  Are there any special considerations that need to be accounted for regarding single, vs multiple collisions of the labelled N2O5 with aerosol and the walls? Or rather, in Fig. 3, how many times would you expect a labelled N2O5 to cycle through a particle or the wall of the flow tube over the time constant of the flow reactor? Is this important to the analysis or derivation of the equations presented?**

*Response:* no it is not important, since we integrate over many labelled molecules arriving in the detection system, and we statistically evaluate in what chemical form they are at the moment of detection. Thus the first order loss rate coefficient is correctly derived from the observed net loss or appearance rates, independent of the number of molecules contributing to these, as long as counting statistics is not limiting the error. In fact, the condensed phase cycling of $N_2O_5$ is very fast, and so is evaporation back to the gas phase, and the number of times an individual molecule (labelled or non-labelled) is entering the particle phase is limited by the collision rate with the particles, which is on the order of one per second for the aerosol conditions used. Since the number density of labelled molecules in the gas phase is of the same order of magnitude as that of the number density of particles, each particle receives at most one labelled $N_2O_5$ molecule during the residence time in the flow tube.

*Action in manuscript:* due to the presentation of a more detailed scheme to showcase the different pathways of labelled and non-labelled $N_2O_5$ in Figure 3a and e, as well as the aqueous phase turnover rates, this aspect is much more clear now.

**3) What is the effect of labelled HNO3 that is generated in the source region?  How would this be interpreted in the experiment.  It is not uncommon for N2O5 sources to be 10:1 HNO3 to N2O5. What is the expected ratio in this experiment?**

*Response:* The citric acid coating used to trap $N_2O_5$ would not be able to differentiate from HNO3, in that sense the detection system is not able to resolve this. Though, the wall loss rate coefficient of labelled HNO3 is so large that for the present flow tube surface to volume ratio the dominant fraction of it would deposit at the entrance of the flow tube (Guimbaud et al., 2002), which was not observed; the uptake coefficient for labelled $N_2O_5$ on the reactor walls was in the range of $10^{-7}$, which would also not lead to substantial $HNO_3$ production.. We will add a note on this in the revised version.

*Action in manuscript:* more details about the impact of HNO3 is given at the beginning of the results and discussion section.

**4) Like reviewer #1, I am also confused by the notation of [N2O5]p. It would be helpful if equation 3 and Figure 2 were consistent in notation. It would also be helpful to denote between labelled and unlabeled here, as the unlabeled N2O5 uptake coefficient to nitrate aerosol could be pretty small even at high labeled N2O5 uptake coefficients, correct?**

*Response:* as already mentioned in the response to Referee #1, we will adapt the notation and make clear that the quantities in equation refer to labelled molecules (which were the only direct observables of the experiment). Yes, the net uptake of non-labelled $N_2O_5$ to nitrate containing aerosol is very low (Table 1, lowest row).

*Action in manuscript:* notation in equation (3) and Figure 2 changed

**Replies to comments for article "Efficient bulk mass accommodation of N2O5 into neutral aqueous aerosol" by G. Gržinić et al.**

Goran Gržinić [1,2], Thorsten Bartels-Rausch[1], Andreas Türler[2,3], Markus Ammann[2]

[1] Laboratory of Environmental Chemistry, Paul Scherrer Institut, Villigen, 5232, Switzerland
[2] Department of Chemistry and Biochemistry, University of Bern, Bern, 3012, Switzerland
[3] Laboratory of Radiochemistry, Paul Scherrer Institut, Villigen, 5232, Switzerland

*Correspondence to*: Markus Ammann (markus.ammann@psi.ch)

**Anonymous Referee #3**

**1. If I understand correctly, the net uptake is determined by measuring the gas-phase 13N and the particle phase 13N, where it is assumed particle phase 13N was only derived from 13N N2O5, and that any 13N nitrate produced from N2O5 stays in the particle phase.**

*Response:* yes, this is correct. Gas phase $^{13}N$ is separated into $N_2O_5$ /$NO_3$ and $NO_2$ respectively, and the contribution of $NO_3$ to the signal is corrected for as mentioned in the text. After removal of gas phase $N_2O_5$ at the entrance of the detection system, only nitrate remains in the particle phase. The approach has been tested with ammonium sulfate (Grzinic et al., 2014) and used in a previous study with citric acid aerosol. The potential occurrence of evaporation of $^{13}N$ labelled $HNO_3$ has been discussed in these previous studies and have been the reason to only work with neutral aerosol in the present study as described in the introduction.

*Action in manuscript:* This is now more explicitly explained towards the end of the experimental section.

**2. Lines 25 on – what are the relative amounts of NO2 and N2O5. NO2 is known to undergo slow disproportionation to HNO3. Is this a concern here? What about wall production of 13HNO3 and its incorporation into the particles due to revolatilization?**

*Response:* the ratio of $NO_2$ to $N_2O_5$ is about 1 at 50% RH and about 0.45 at 70% RH. In the dry $N_2O_5$ online production flow tube, $NO_2$ losses (due to heterogeneous disproportionation) are undetectable ($<10^{-8}$). Disproportionation of $NO_2$ in the aerosol phase is negligible. As mentioned by Referee #2, $HNO_3$ would rather be of concern as a product of hydrolysis of $N_2O_5$ on the walls, for which we did not have evidence to impact on the measurements (see our response there). We have been more concerned by evaporation of non-labelled $HNO_3$ from the walls that might lead to additional acidification and may thus contribute to the evaporation of labelled $HNO_3$ product from acidic aerosol discussed in our previous studies (Grzinic et al., 2014). But given the $k_w$ values reported, and if assuming that all $N_2O_5$ lost to the walls would lead to $HNO_3$ in the gas phase and that $HNO_3$ would be taken up with unit uptake coefficient, particle phase nitrate and acidity remain dominated by that deriving from direct $N_2O_5$ uptake.

*Action in manuscript:* The effects of HNO3 are now discussed in more detail at the beginning of the results and discussion section.

**3. I'm not sure that the net reaction probabilities and their interpretation is correct or at least well explained. I can see why the net uptake measured in this 13N system on nitrate containing particles would be higher than that for unlabeled N2O5 on nitrate containing particles, due to the added channel of "isotope exchange" (for short). But I do not see why there would be an order of magnitude larger net reactive uptake of 13N N2O5 on nitrate containing particles compared to 13N N2O5 reacting on sulfate particles. I think a simple box model using the ratio of k5/k7 and nitrate and water molalities might help.**

*Response:* as mentioned above we will show some box model simulations in the revised version to show the different behaviors of labelled and non-labelled $N_2O_5$ in presence and absence of nitrate. Essentially, in absence of nitrate, labelled $N_2O_5$ behaves as non-labelled $N_2O_5$, because the amount of non-labelled nitrate in the particle phase is not enough for the 'isotope exchange' to become effective. In turn, it remains difficult to fully constrain rate coefficients for the different elementary reaction based on the limited dataset only.

*Action in manuscript:* Box model calculations are supplied and explained in the text in detail, along with the comparison between the cases of nitrate containing aerosol and pure sulfate. The isotope exchange becomes much more apparent, also from the more detailed reaction scheme provided in Figure 3, accompanying the box model results.

**4. Or the description of the results and their interpretation needs to be clearer. I don't see where the limitation in the sulfate system is to produce that much lower of a net reaction probability, if there is equal probability of 13N N2O5 becoming 13NO3- and 13NO2+. What would be the limitation to 13NO2+ in the sulfate particles that doesn't exist in the nitrate particles?**

*Response:* as mentioned just above, $^{13}NO_2^+$ simply reacts with non-labelled nitrate back to $N_2O_5$ if not reacted with $H_2O$ to form labelled nitrate. Therefore, on sulfate, the labelled N has the same fate as non-labelled N.

*Action in manuscript:* as mentioned above the comparison between the two cases is now explicitly explained in plots of the box model calculation results and the reaction scheme.

**5. Lines 10 onward, page 2 – I don't think all previous estimates of bulk accommodation were as low as implied here, only that net reactive uptake was lower. In other papers, a typical assumption was also that the accommodation coefficient was not limiting net uptake, see for e.g. Bertram and Thornton ACP 2009 and others.**

*Response:* ok, we will detail this a bit more to mention the various limiting processes (dissociation, accommodation) and their ranges that have been applied to explain different datasets.

*Action in manuscript:* An explicit discussion about the different limiting processes is now given.

**6. Line 23, page 5 – sentence that starts here as a double negative at the end, making it difficult to interpret the meaning. Suggest: Thus, in this branch, all 13N labels remain in the aqueous phase given that HNO3 evaporation from neutral aerosol is unlikely.**

*Response:* ok, thank you.

*Action in manuscript:* text changed accordingly.

**7. Line 7, page 6 – this paragraph is interesting. In the Bertram and Thornton parameterization, there is a dependence of the equilibrium between N2O5(aq) = NO3-(aq) + NO2+(aq) on the water activity (molar concentration in that paper) that was inferred from the behavior of observed uptake coefficients. This dependence was in addition to that inferred for the fate of the NO2+. Seems this presents a possible physical explanation.**

*Response:* Thank you for this hint; we are going into substantially more detail in the revised version to discuss the different kinetic assumptions in different suggested parameterisations and schemes, along with the box model calculations. This then also includes more details with respect to the dependence on water activity.

*Action in manuscript:* in addition to the extended discussion, we also reiterate the point of salting effects and the different kinetic regime towards the end of the results and discussion section.

**8. Conclusions, last sentence: "The absence of accommodation limitation also helps rationalizing results from field experiments (Phillips et al., 2016; Wagner et al., 2013), where under some conditions uptake coefficients constrained by combined N2O5 and aerosol measurements are larger than those suggested by laboratory studies available to date."**

**I'm not sure I completely agree with this conclusion. Do the measurements presented herein actually help explain field determinations of the reaction probability being larger than those measured in the laboratory? A few lines above, it was noted that the net reaction probability from this set of measurements was similar to previous values (at least for sulfate particles). The field observations are not deriving accommodation coefficients, but net reaction probabilities. I agree that if some previous laboratory studies concluded mass accommodation coefficients were <0.04, then field observations of net reaction probability greater than 0.04 would not be supported by such a conclusion (or vice versa).**

*Response:* we agree, our point is maybe a bit too far going. We wanted to express the fact that most likely bulk accommodation would not limit uptake; and if strong sinks for $NO_2^+$ would be present, accommodation would not limit reaction. We also agree that uptake coefficients inferred from field experiments larger than those in laboratory studies are not that frequent

and often remain uncertain. We will adapt the tonality of our conclusions to reflect this discussion.

*Action in manuscript:* Throughout the text, we have more clearly differentiated between nitrate containing aerosol to which our experiments relate, and non- (or less) nitrate containing aerosol, for which we can only suggest similar accommodation and dissociation rates. We have dropped the explicit comparison to the two field studies.

**However, other laboratory studies and even parameterizations for similar aerosol systems as those used here, have concluded or assumed mass accommodation is not limiting (e.g. Bertram and Thornton, ACP 2009 and references therein), and thus that the limitations are in the chemistry and diffusion.**

*Response:* We will mention the different limitations invoked in the different studies in more detail, and yes, we agree that physical effects (solubility and diffusion) might often be more important than considered so far.

*Action in manuscript:* as mentioned above, these physical effects are discussed in more detail now.

**Thus, I would argue that for the reaction probabilities derived from field measurements that are larger than some laboratory measurements (pretty small fraction) are because of three possible reasons: (1) laboratory measurements have an additional limitation to net uptake not present (or reduced) in the atmosphere (could be salting out as suggested), (2) there are additional faster reaction pathways for N2O5 in ambient aerosol particles not yet probed in the laboratory, or (3) because the assumptions required in deriving the reaction probabilities from field observations are flawed.**

**It is hard to imagine there is something that reacts with NO2+ much faster than Chloride in solution (though it is possible), and in aqueous sea salt particles, the reaction probability is still only measured to be similar to that on aqueous sulfate particles. That to me suggest the limitation that seems to keep reaction probabilities < 0.05 at room temperature is less likely the NO2+ chemistry and instead the solubility, dissociation, and diffusion beforehand. Anyway, that is a long winded way of saying that I don't think there is support for that conclusion as written.**

*Response:* we agree, we will summarize this discussion in the conclusion to reduce the potential impact of knowing that bulk accommodation is fast. The study remains certainly valuable enough because it provides direct evidence for the elementary reactions underlying aqueous phase $N_2O_5$ chemistry and also allows constraining their rate coefficients at least for an aqueous aerosol with high nitrate content. We will adapt the tonality of our conclusions to reflect this discussion.

*Action in manuscript:* as mentioned above, we have now avoided a too detailed discussion of field experiments and their implication, but more explicitly constrain the conclusions to the aerosol compositions actually studied.

**Replies to comments for article "Efficient bulk mass accommodation of N2O5 into neutral aqueous aerosol" by G. Gržinić et al.**

Goran Gržinić [1,2], Thorsten Bartels-Rausch[1], Andreas Türler[2,3], Markus Ammann[2]

[1] Laboratory of Environmental Chemistry, Paul Scherrer Institut, Villigen, 5232, Switzerland
[2] Department of Chemistry and Biochemistry, University of Bern, Bern, 3012, Switzerland
[3] Laboratory of Radiochemistry, Paul Scherrer Institut, Villigen, 5232, Switzerland

*Correspondence to*: Markus Ammann (markus.ammann@psi.ch)

**Anonymous Referee #4**

**1. I'd like the authors to include an estimate of the mixing ratio of N2O5 in the flow tube, as well as HNO3, as well as Va and Sa values (needed in the diffusion correction and to establish the role of possible aerosol size effects in the uptake process).**

*Response:* The aerosol surface area density has already been given in the original table, and the average diameter is mentioned in the text along with the discussion of the diffusion correction.

As mentioned in the reply to question 3 from Referee #2, and question 2 from Referee #3, the design of the experimental setup does not allow to directly assess the levels of $HNO_3$. As discussed there, for the configuration used in the present setup, we would have lost labelled $HNO_3$ to the walls efficiently, and based on the measured wall loss rates, the non-labelled $HNO_3$ levels building up as product would remain low enough to not significantly affect nitrate levels or acidity in the aerosol phase. The gas-phase non-labeled $NO_2$ and $N_2O_5$ concentrations were around $10^{11}$ molecule $cm^{-3}$. Both aspects will be explicitly mentioned in the revised version.

*Action in manuscript:* Effects of $HNO_3$ are discussed in more detail at the beginning of the results and discussion section.

**2. The results hang together well but the low uptake onto sodium sulfate aerosol needs to be discussed in more detail. In particular, the question needs to be addressed as to why is it so low - lower than the reported value by Wahner et al. of 0.037 (+ 0.008 - 0.019)?**

*Response:* As mentioned in the response to the question below, crystallization of the $Na_2SO_4$ aerosol at 50% is the most likely explanation for the low uptake value measured. The value reported in Mentel et al (1999), which was measured at 71% RH, is consistent with our measurement at 70%. However the measurement done at 50% RH is much closer to the values reported in Hallquist et al. (2003) for dry $(NH_4)_2SO_4$ aerosol. This will be added in the revised version.

*Action in manuscript:* This has been added to the results and discussion section.

**3. The aerosol is dried to 40% RH ('to prevent effloresence' line 39, page 2). This is quite far below the reported efflorescence RH of Na2SO4 of 58%, (e.g. quoted in J. Phys. Chem. A, 2007, 111 (42), pp 10660–10666), so partial or complete efflorescence may explain the low values of uptake observed for the 50% RH case of Na2SO4, as uptake onto solid sulfate is known to be slower. Could this be a possible reason?**

*Response:* Indeed, partial or complete crystallization seems like the most likely explanation, as mentioned by the reviewer, and in light of what is reported by Gao et al. 2007 with regards to the efflorescence relative humidity for Na2SO4. We will update the paper to reflect this.

*Action in manuscript:* As mentioned above, this has been added to the results and discussion section.

**4. I think the discussion of salting effects requires more detail (do they have any evidence for the salting they propose?) and to be better linked to variation in RH.**

*Response:* within the limited data set of this study, it is difficult to become more conclusive about the salting effects. As mentioned in the discussion, on page 6, from line 7, the fact that the water activity had a stronger effect on the uptake coefficient than different (though still high) nitrate to sulfate molar ratios. As indicated above in response to referee #3, we could only add that Bertram and Thornton (2009) have seen independent evidence for a water activity dependent equilibrium. Also the application of a more detailed kinetic scheme in box model calculations shown in the revised version does not allow constraining the solubility further.

*Action in manuscript:* This discussion has been added towards the end of the discussion section when reiterating the physical effects on the kinetic regime.

**References**

[revised manuscript text omitted]

**Description of box model calculations**

The box model calculations are explicitly treating $N_2O_5$ chemistry in the aqueous phase for both labelled and non-labelled $N_2O_5$ molecules, as well as exchange of both with the gas phase. Diffusion is neglected in both phases. Labelling of reactions is done as in main text and as indicated in Figure 3a of the main text.

The following chemical reactions have been used to describe the aqueous phase chemistry of non-labelled $N_2O_5$:

$$N_2O_{5(aq)} \longrightarrow NO_{2(aq)}^+ + NO_{3(aq)}^- \qquad k_4 \qquad (R4)$$

$$NO_{2(aq)}^+ + NO_{3(aq)}^- \longrightarrow N_2O_{5(aq)} \qquad k_5 \qquad (R5)$$

$$NO_{2(aq)}^+ + H_2O_{(liq)} \longrightarrow NO_{3(aq)}^- + 2H_{(aq)}^+ \qquad k_7 \qquad (R7)$$

$$N_2O_{5(g)} \underset{\longleftarrow}{\overset{\longrightarrow}{}} N_2O_{5(aq)} \qquad k_{gbg}, k_{gbb}, k_{bgb}, k_{bgg} \qquad (R3)$$

The chemical scheme for labelled $N_2O_5$ is analogous and assumes that the position of the labelled $N^*$ atom within the $N_2O_5$ is not relevant, such that upon dissociation, labelled nitronium and labelled nitrate are formed at equal rates, but that otherwise the rate coefficients are the same as for the non-labelled species:

$$N^*NO_{5(aq)} \longrightarrow N^*O^+_{2(aq)} + NO^-_{3(aq)} \qquad k_4/2 \qquad\qquad \text{(R4')}$$

$$N^*NO_{5(aq)} \longrightarrow NO^+_{2(aq)} + N^*O^-_{3(aq)} \qquad k_4/2 \qquad\qquad \text{(R4'')}$$

$$N^*O^+_{2(aq)} + NO^-_{3(aq)} \longrightarrow N^*NO_{5(aq)} \qquad k_5 \qquad\qquad \text{(R5')}$$

$$NO^+_{2(aq)} + NO^-_{3(aq)} \longrightarrow NNO_{5(aq)} \qquad k_5 \qquad\qquad \text{(R5)}$$

$$NO^+_{2(aq)} + N^*O^-_{3(aq)} \longrightarrow N^*NO_{5(aq)} \qquad k_5 \qquad\qquad \text{(R5'')}$$

$$N^*O^+_{2(aq)} + H_2O_{(liq)} \longrightarrow N^*O^-_{3(aq)} + 2H^+_{(aq)} \qquad k_7 \qquad\qquad \text{(R7')}$$

$$NO^+_{2(aq)} + H_2O_{(liq)} \longrightarrow NO^-_{3(aq)} + 2H^+_{(aq)} \qquad k_7 \qquad\qquad \text{(R7)}$$

$$N^*NO_{5(g)} \underset{\longleftarrow}{\longrightarrow} N^*NO_{5(aq)} \qquad k_{gbg}, k_{gbb}, k_{bgb}, k_{bgg} \qquad \text{(R3')}$$

$$N_2O_{5(g)} \underset{\longleftarrow}{\longrightarrow} N_2O_{5(aq)} \qquad k_{gbg}, k_{gbb}, k_{bgb}, k_{bgg} \qquad \text{(R3)}$$

Due to the small fraction of labelled N, reactions among labelled species are not included in the scheme. The set of differential equations listed in Table S1 were used to solve for the evolution of all gas and aqueous phase species with time using Matlab.

The exchange rate coefficients to describe gas-aqueous and aqueous-gas transfer of $N_2O_5$ were obtained from the steady state condition given by Henry's law equilibrium, thus gas – aerosol equilibrium in absence of chemical reaction (assuming that the gas phase chemical reactions of $N_2O_5$ are in steady state on the time scale of gas – aerosol equilibration). The steady state of the gas phase $N_2O_5$ concentration in equilibrium with aqueous phase $N_2O_5$ is described by:

$$\frac{d[N_2O_5]_g}{dt} = k_{bg}[N_2O_5]_{aq}V_p N_{av} - \frac{\alpha_b \omega}{4}S_p[N_2O_5]_g = 0 \Rightarrow k_{bg} = \frac{\alpha_b \omega S_p}{4V_p N_{av}H'}$$

$$H' = \frac{[N_2O_5]_{aq}}{[N_2O_5]_g} = \frac{[N_2O_5]_{aq}}{N_{av}p_{N2O5}/RT} = H\frac{RT}{N_{av}}$$

$k_{bg}$ is the first order rate coefficient for the release to the gas phase in units of $s^{-1}$, $V_p$ is the total aqueous phase aerosol volume per volume of gas phase in liter $cm^{-3}$, $S_p$ the total aerosol surface area per volume of gas phase in $cm^2\ cm^{-3}$, $p_{N2O5}$ is the partial pressure of $N_2O_5$ in atm,

$R$ is the gas constant (8.314 J mole$^{-1}$ K$^{-1}$), $T$ is the temperature, $H$ is the Henry's law constant in M atm$^{-1}$, $H'$ is the ratio of the aqueous phase molarity of N$_2$O$_5$ to the gas phase concentration in molecule cm$^{-3}$ and $N_{av}$ is Avogadro's number (note that [N$_2$O$_5$]$_{aq}$ is in units of M, while [N$_2$O$_5$]$_g$ is in units of molecule cm$^{-3}$). The same result can also be obtained from considering steady state for the aqueous phase N$_2$O$_5$ concentration in an individual aerosol particle:

$$\frac{d[N_2O_5]_{aq}}{dt} = -k_{bg}[N_2O_5]_{aq} + \frac{\alpha_b \omega s_p}{4v_p N_{av}}[N_2O_5]_g = 0 \Rightarrow k_{bg} = \frac{\alpha_b \omega s_p}{4v_p N_{av}H'} \quad \frac{s_p}{v_p} = \frac{S_p}{V_p}$$

Where $v_p$ is the volume of an individual aerosol particle in liter, and $s_p$ its surface area in cm$^2$. Thus, the rate coefficients describing the gas – aqueous phase exchange are:

$$k_{gbb} = \frac{\alpha_b \omega s_p}{4v_p N_{av}} \qquad \left[\frac{\text{mole cm}^3}{\text{s liter molecule}}\right]$$

$$k_{gbg} = \frac{\alpha_b \omega S_p}{4} \qquad [\text{s}^{-1}]$$

$$k_{bgb} = \frac{\alpha_b \omega s_p}{4v_p N_{av}H'} \qquad [\text{s}^{-1}]$$

$$k_{bgg} = \frac{\alpha_b \omega S_p}{4H'} \qquad \left[\frac{\text{molecule liter}}{\text{s cm}^3 \text{ mole}}\right]$$

The apparent net uptake coefficient, $\gamma$ and $\gamma^*$, can be derived from the effective loss rates of the gas phase N$_2$O$_5$ and N$^*$NO$_5$, respectively, obtained from the solutions to the differential equations, via:

$$\gamma = \left[-\frac{d[N_2O_5]_g}{dt}\right] \bigg/ \left[\frac{\omega}{4}S_p[N_2O_5]_g\right] = \left[k_{bgg}[N_2O_5]_{aq} - k_{gbg}[N_2O_5]_g\right] \bigg/ \left[\frac{\omega}{4}S_p[N_2O_5]_g\right]$$

$$\gamma^* = \left[-\frac{d[N^*NO_5]_g}{dt}\right] \bigg/ \left[\frac{\omega}{4}S_p[N^*NO_5]_g\right] = \left[k_{bgg}[N^*NO_5]_{aq} - k_{gbg}[N^*NO_5]_g\right] \bigg/ \left[\frac{\omega}{4}S_p[N^*NO_5]_g\right]$$

We note that due the fact that we neglect aqueous phase diffusion in the kinetics, these uptake coefficients are only representing estimates and should be compared with caution to the measured data. In principle they represent the uptake coefficients expected for small enough particles (smaller than those in the experiments) such that liquid phase diffusion would not play a role. While the apparent reacto-diffusive length based on the net hydrolysis kinetics is rather large, the true dissociation as suggested here is rather fast. Inclusion of diffusion would require an explicit depth resolving model, which was beyond the scope of this analysis. The

purpose of the simulations here was to simulate the behavior of labelled and non-labelled $N_2O_5$ in the aqueous phase and especially the exchange of labelled nitrate with the non-labelled nitrate pool for the case of $NaNO_3$.

For the simulations of the $NaNO_3$ and mixed $Na_2SO_4$ / $NaNO_3$ cases, we used the parameters for the kinetics as compiled by Mentel et al. (1999). The aerosol composition was derived from the E-AIM model as described in the main text and listed there in Table 1. Mentel et al. (1999) assumed the solubility of $N_2O_5$ to be 5 M atm$^{-1}$ (2 M atm$^{-1}$ in the IUPAC recommended parameterization). As mentioned in the main text and in the other recent studies (Bertram and Thornton, 2009), in view of the fact that the concentration of nitronium has never been measured, the solubility and also the other rate coefficients determining its concentration are not well constrained. Mentel et al. set the dissociation rate coefficient, $k_1$, to $5\times10^6$ s$^{-1}$, the rate coefficient for the recombination, $k_2$, to $2.4\times10^{10}$ M$^{-1}$ s$^{-1}$, and the first rate coefficient for the reaction of nitronium with water, $k_3\times[H_2O]_{aq}$, to $\times10^9$ s$^{-1}$ (note that the values given in Mentel et al.'s Table 2 are on a molality unit basis). This set of parameters that led to agreement with their own data and also nicely explain the data with labelled $N_2O_5$ presented in this study in the main text without further adjustments.

For the simulations of the $Na_2SO_4$ case, the dissociation and recombination rate coefficients were kept constant, but the rate of reaction of nitronium with water, $k_3\times[H_2O]_{aq}$, was reduced by one order of magnitude to $\times10^9$ s$^{-1}$, to obtain net uptake coefficients in agreement with the measurement. As mentioned and discussed in the main text, reducing the dissociation rate would have a similar effect. However, the parallel behavior of labelled and non-labelled $N_2O_5$ for the $Na_2SO_4$ aerosol is not affected by that.

**Table S1:** Differential equations. $[N_2O_5]_{aq}$ is in units of M, while $[N_2O_5]_g$ is in units of molecule cm$^{-3}$. The rate coefficients are explained in more detail below.

| | |
|---|---|
| $$\frac{d[N_2O_5]_{aq}}{dt} = -k_1[N_2O_5]_{aq} + k_2[NO_2^+]_{aq}[NO_3^-]_{aq} - k_{bgb}[N_2O_5]_{aq} + k_{gbb}[N_2O_5]_g$$ | D1 |
| $$\frac{d[N^*NO_5]_{aq}}{dt} = -k_1[N^*NO_5]_{aq} + k_2[N^*O_2^+]_{aq}[NO_3^-]_{aq} + k_2[NO_2^+]_{aq}[N^*O_3^-]_{aq} - k_{bgb}[N^*NO_5]_{aq} + k_{gbb}[N^*NO_5]_g$$ | D2 |
| $$\frac{d[NO_2^+]_{aq}}{dt} = k_1[N_2O_5]_{aq} + \frac{k_1}{2}[N^*NO_5]_{aq} - k_2[NO_2^+]_{aq}[NO_3^-]_{aq} - k_2[NO_2^+]_{aq}[N^*O_3^-]_{aq} - k_3[NO_2^+]_{aq}[H_2O]_{aq}$$ | D3 |
| $$\frac{d[N^*O_2^+]_{aq}}{dt} = \frac{k_1}{2}[N^*NO_5]_{aq} - k_2[N^*O_2^+]_{aq}[NO_3^-]_{aq} - k_3[N^*O_2^+]_{aq}[H_2O]_{aq}$$ | D4 |
| $$\frac{d[NO_3^-]_{aq}}{dt} = k_1[N_2O_5]_{aq} + \frac{k_1}{2}[N^*NO_5]_{aq} - k_2[NO_2^+]_{aq}[NO_3^-]_{aq} - k_2[N^*O_2^+]_{aq}[NO_3^-]_{aq} + k_3[NO_2^+]_{aq}[H_2O]$$ | D5 |
| $$\frac{d[N^*O_3^-]_{aq}}{dt} = \frac{k_1}{2}[N^*NO_5]_{aq} - k_2[NO_2^+]_{aq}[N^*O_3^-]_{aq} + k_3[N^*O_2^+]_{aq}[H_2O]_{aq}$$ | D6 |
| $$\frac{d[H_2O]_{aq}}{dt} = -k_3[N^*O_2^+]_{aq}[H_2O]_{aq} - k_3[NO_2^+]_{aq}[H_2O]_{aq}$$ | D7 |
| $$\frac{d[H^+]_{aq}}{dt} = 2k_3[N^*O_2^+]_{aq}[H_2O]_{aq} + 2k_3[NO_2^+]_{aq}[H_2O]_{aq}$$ | D8 |
| $$\frac{d[N_2O_5]_g}{dt} = k_{bgg}[N_2O_5]_{aq} - k_{gbg}[N_2O_5]_g$$ | D9 |
| $$\frac{d[N^*NO_5]_g}{dt} = k_{bgg}[N^*NO_5]_{aq} - k_{gbg}[N^*NO_5]_g$$ | D10 |

**Table S2.** Data for $NaNO_3$, $Na_2SO_4$ and 1:1 $NaNO_3/Na_2SO_4$ aerosol experiments shown in Figure 2 of the main text.

| $NaNO_3$ | | | | | |
|---|---|---|---|---|---|
| 50% RH | | | 70% RH | | |
| Time (s) | $C^*_p(t) / C^*_p (t=0)$ | std.dev. | Time (s) | $C^*_p(t) / C^*_p (t=0)$ | std.dev. |
| 0.75 | 1.78E-02 | 3.63E-03 | 0.75 | 4.17E-02 | 6.74E-03 |
| 1.26 | 2.73E-02 | 5.30E-03 | 1.25 | 6.40E-02 | 7.70E-03 |
| 1.76 | 2.83E-02 | 3.91E-03 | 1.75 | 8.61E-02 | 8.49E-03 |
| 2.27 | 3.62E-02 | 4.58E-03 | 2.25 | 1.07E-01 | 8.33E-03 |
| 2.77 | 5.25E-02 | 5.94E-03 | 2.76 | 1.17E-01 | 1.12E-02 |
| 3.28 | 6.12E-02 | 4.09E-03 | 3.26 | 1.48E-01 | 1.10E-02 |
| 3.78 | 6.80E-02 | 5.59E-03 | 3.76 | 1.68E-01 | 1.49E-02 |
| $Na_2SO_4$ | | | | | |
| 51% RH | | | 70% RH | | |
| Time (s) | $[N_2O_5]_p(t) / [N_2O_5]_g (t=0)$ | std.dev. | Time (s) | $[N_2O_5]_p(t) / [N_2O_5]_g (t=0)$ | std.dev. |
| 0.76 | 1.18E-03 | 2.65E-03 | 0.75 | 4.70E-03 | 3.48E-03 |
| 1.26 | 1.57E-03 | 1.61E-03 | 1.25 | 5.44E-03 | 4.31E-03 |
| 2.79 | 2.63E-03 | 2.61E-03 | 1.75 | 6.52E-03 | 3.32E-03 |
| | | | 2.25 | 8.86E-03 | 3.45E-03 |
| | | | 2.76 | 1.19E-02 | 3.36E-03 |
| $NaNO_3+Na_2SO_4$ | | | | | |
| 51% RH | | | 70% RH | | |
| Time (s) | $[N_2O_5]_p(t) / [N_2O_5]_g (t=0)$ | std.dev. | Time (s) | $[N_2O_5]_p(t) / [N_2O_5]_g (t=0)$ | std.dev. |
| 0.75 | 1.27E-02 | 2.14E-03 | 0.75 | 3.33E-02 | 4.19E-03 |
| 1.25 | 1.55E-02 | 3.26E-03 | 1.25 | 5.95E-02 | 8.93E-03 |
| 1.75 | 2.50E-02 | 3.65E-03 | 1.75 | 6.92E-02 | 1.13E-02 |
| 2.25 | 2.06E-02 | 3.47E-03 | 2.25 | 1.01E-01 | 1.21E-02 |
| 2.76 | 2.45E-02 | 3.99E-03 | 2.76 | 1.14E-01 | 8.94E-03 |
| 3.26 | 2.74E-02 | 3.47E-03 | | | |
| 3.76 | 3.92E-02 | 3.61E-03 | | | |